# Concomitant Chemoradiation Therapy with Gold Nanoparticles and Platinum Drugs Co-Encapsulated in Liposomes

**DOI:** 10.3390/ijms21144848

**Published:** 2020-07-09

**Authors:** Gabriel Charest, Thititip Tippayamontri, Minghan Shi, Mohamed Wehbe, Malathi Anantha, Marcel Bally, Léon Sanche

**Affiliations:** 1Department of Nuclear Medicine and Radiobiology and Medical Research Center, Faculty of Medicine and Health Sciences, Université de Sherbrooke, Sherbrooke, QC J1H 5N4, Canada; Thititip.T@chula.ac.th (T.T.); Minghan.Shi@USherbrooke.ca (M.S.); Leon.Sanche@usherbrooke.ca (L.S.); 2Department of Radiological Technology and Medical Physics, Chulalongkorn University, Bangkok 10330, Thailand; 3British Columbia Cancer Agency (BCCA), Vancouver, BC V6H 3Z6, Canada; mwehbe@bccrc.ca (M.W.); manantha@bccrc.ca (M.A.); mbally@bccrc.ca (M.B.)

**Keywords:** radiosensitization, radioenhancement, liposomes, gold nanoparticles, platinum drugs, low energy electrons, chemoradiotherapy, cancer treatment

## Abstract

A liposomal formulation of gold nanoparticles (GNPs) and carboplatin, named LipoGold, was produced with the staggered herringbone microfluidic method. The radiosensitizing potential of LipoGold and similar concentrations of non-liposomal GNPs, carboplatin and oxaliplatin was evaluated in vitro with the human colorectal cancer cell line HCT116 in a clonogenic assay. Progression of HCT116 tumor implanted subcutaneously in NU/NU mice was monitored after an irradiation of 10 Gy combined with either LipoGold, GNPs or carboplatin injected directly into the tumor by convection-enhanced delivery. Radiosensitization by GNPs alone or carboplatin alone was observed only at high concentrations of these compounds. Furthermore, low doses of carboplatin alone or a combination of carboplatin and GNPs did not engender radiosensitization. However, the same low doses of carboplatin and GNPs administered simultaneously by encapsulation in liposomal nanocarriers (LipoGold) led to radiosensitization and efficient control of cell proliferation. Our study shows that the radiosensitizing effect of a combination of carboplatin and GNPs is remarkably more efficient when both compounds are simultaneously delivered to the tumor cells using a liposomal carrier.

## 1. Introduction

The necessity to protect healthy tissues surrounding a tumor is a major issue in radiotherapy and limits the therapeutic radiation dose. A promising avenue to increase this dose, while decreasing collateral damage consists of developing radiosensitizers or radio-enhancers that increase relative biological effectiveness (RBE) at the tumor [1,2]. In principle, such treatment could be optimized by injecting the radiosensitizer directly into the tumor together with a chemotherapeutic agent. In this case, the detrimental effects of increased radiation dose and chemotherapy would be much more localized at the tumor. Based on recent advances in molecular radiation physics, we show in this article that such a combination may offer an interesting alternative application of chemoradiation therapy. We suggest encapsulating in liposomes both a gold nanoparticle (GNP) to enhance the local dose and a Pt drug to act as a chemotherapeutic agent (CA). The GNPs supply a high density of short-range low energy electrons (LEEs), which strongly and locally react with the Pt-CA bound to vital biomolecules, such as DNA.

When X-rays interact with a gold nanoparticle (GNP), LEEs are emitted in large numbers [3,4,5]. The electron emission coefficient is larger by roughly two orders of magnitude in the 10–80 keV range, compared to that of biological tissue [6]. The electron energy distribution extends from a few eV to energies close to the primary photon energy [4,7,8]. Considering the range of electrons in water and the spherical geometrical factor, most of the LEE density lies within submicrometer distances from the surface of a GNP irradiated with such X-rays [9]. Electrons with energies above the ionization potential of the medium produce a further generation of LEEs. Electrons of energy below 30 eV (i.e., LEEs) are the most numerous reactive species created around the GNP and carry a large portion of the energy absorbed by the metal [7,8,10]. Thus, GNPs are essential generators of LEEs, which have extremely short ranges and can confine, within the nanometer range, damages created by ionizing radiation (IR). Furthermore, a different mechanism of radiolysis, related to the production of hydroxyl radicals in the presence of GNPs in solution, has been shown to contribute to their radiosensitizing properties [11].

LEEs strongly react with the surrounding medium, mainly via a resonant interaction, which leads to their temporary capture by a molecule or a moiety of a large biomolecule, near the GNP. This leads to the formation of a molecular transient anions (TAs) [12]. TAs can decay by reemitting the captured electron (i.e., autoionization) or by dissociation (i.e., dissociative electron attachment). Since autoionization can leave the molecule in a dissociative electronic state, both channels can efficiently break bounds and damage biomolecules [12]. The type of damages induce by LEEs in DNA has been analyzed in detail by various techniques. It includes, base damage and cleavage, single and double strand breaks and other clustered lesions, consisting of strand breaks and base damages [11,13,14,15,16]. Similar damages were observed and enhanced when GNPs were bound to DNA [17,18,19,20].

Recent investigations of the action of LEEs with the types of Pt-DNA complexes formed during radiotherapy in the nucleus of cells have demonstrated that the platinum-based chemotherapeutic agents (Pt-CAs) cisplatin, oxaliplatin, and carboplatin increase two- to three-fold the damage induced by LEEs, causing higher levels of potentially lethal lesions including cluster damages [21,22]. Furthermore, the energy required for LEEs to induce cluster damage to DNA was found to be much lower when the Pt-CAs were bound to DNA [22].

The enhancement of radiation damage and the biological effects, associated with the individual physical and chemical properties of Pt-CAs and GNPs, have been observed in several in vitro and in vivo studies [23,24,25,26,27,28,29,30,31,32,33,34,35,36]. However, from the results obtained individually with GNPs and Pt-CAs, it became obvious that combining both radiosensitizers in a common carrier could considerably enhance the benefit of chemoradiation therapy (i.e., GNPs produce large quantities of LEEs that would strongly react with any Pt-CAs bound to DNA or other biomolecules vital for cell survival). When Pt-CAs or GNPs were bound separately to DNA, high-energy radiation damage was found to increase by factors of 2–4 [18,19,20]. However, when both GNPs and Pt-CAs were bound to DNA and subjected to the same experimental conditions, DNA double strand breaks increased by a factor of 7.5 [37]. This last observation led to the present in vitro and in vivo experiments to evaluate the anti-cancer potential of this combination. In the present study, we assess the synergy between 80-keV X-ray irradiation and GNPs administered together with Pt-CAs to cancer cells and malignant tumors. To increase the probability of local combination of GNPs and Pt-CAs within tumoral tissue, we produced a liposomal vehicle named LipoGold that can efficiently deliver both compounds simultaneously to tumor cells. The HCT116 human colorectal cancer cell line was used for in vitro clonogenic experiments with carboplatin and oxaliplatin as the CA. Tumor response was determined in immunodeficient NU/NU nude mice with carboplatin as the CA. According to recent results, the limitation of radiotherapy in colorectal cancer treatment [25] could be improved through the use of a liposomal formulation carrying both a CA and radio enhancer. Preliminary results are concisely presented in this article on this novel method of combining metal nanoparticles with CAs in an effort to enhance the benefits of concomitant chemoradiation therapy. They are provided for various combinations of CA, GNP, and dose of X-ray radiation.

## 2. Results and Discussions

### 2.1. Clonogenic Assays

Figure 1 shows the cytotoxicity of HCT116 cells for five different treatments (carboplatin, GNP, GNP + carboplatin, LipoGold, oxaliplatin and GNP + oxaliplatin) at different relative concentrations, with or without a fixed X-ray dose of 2 Gy. The mean survival fraction (SF) for all groups are relative to the control group (CTRL, column 3), which is normalized to 1. For the groups treated without irradiation: The low dose carboplatin [0.144 µg/mL] group (column 1) and the combination of GNPs [1.106 µg/mL] plus carboplatin [0.144 µg/mL] group (column 2) have a SF higher than the control group (CTRL, column 3) LipoGold (column 4), that contains the same concentrations as the bare products (GNPs [1.106 µg/mL] and carboplatin [0.144 µg/mL] (column 2)), produces a SF barely lower than that of the CTRL (*p* = 0.06). Groups for oxaliplatin [0.3973 µg/mL] (column 5), a high dose of GNPs [100 µg/mL] (column 7), carboplatin [18.262 µg/mL] (column 10) and the combination of GNPs [100 µg/mL] plus oxaliplatin [0.3973 µg/mL] (column 11), have SF between 0.61 and 0.40. The combination of a high dose of GNPs [100 µg/mL] plus carboplatin [18.262 µg/mL] (column 12) has a SF of 0.29.

The groups treated with IR (i.e., the low dose of carboplatin group [0.144 µg/mL] (column 6) and the combination of GNPs [1.106 µg/mL] plus carboplatin [0.144 µg/mL] group (column 8)) have a mean SF, which are slightly higher (but not significantly) than the group treated with radiation alone (column 9). Groups have a SF between 0.27 and 0.22 include GNPs [100 µg/mL] (column 13), oxaliplatin [0.3973 µg/mL] (column 14) and LipoGold (column 15). SF lower than 0.14 were observed for carboplatin [18.262 µg/mL] (column 16), GNP [100 µg/mL] plus carboplatin [18.262 µg/mL] column 17) and GNP [100 µg/mL] plus oxaliplatin [0.3973 µg/mL] (column 18).

The clonogenic assays (Figure 1) of HCT116 treated with oxaliplatin agrees well with a previous study [38]. Surprisingly, the groups treated with low dose of carboplatin (column 1) or combination of low dose of GNP + carboplatin (column 2) have a higher, but not significant, SF compared to the control group (column 3). A similar ordering is found when IR is combined with these treatments (columns 6, 8) relative to IR alone (column 9). These results suggest that these compounds, at non-lethal doses, could trigger the expression of the repair system that promotes cell survival. However, higher doses of the same compounds without IR (columns 10 and 12) or with IR (columns 16 and 17) do reduce considerably and significantly the SF. Interestingly, when a low dose of GNP + carboplatin are encapsulated in the liposomal formulation LipoGold (column 4), a significant lower SF is observed compared to the free compounds (column 2, *p* = 0.046) and but was not significant compared to the control (column 3, *p* = 0.061). The same pattern is observed when LipoGold is combined with IR (column 15 compared to columns 8 and 9). *The most noteworthy result from this* in vitro *experiment is certainly the demonstration that simultaneous delivery of the compounds by the liposomal formulation is more efficient than the same amount of these compounds without liposomal carrier*. This is clearly seen (annotated in Figure 1) without radiation (column 4 vs. column 2) and with radiation (column 15 vs. column 8).

The stimulation of cell proliferation by a low dose of carboplatin is suspected to be an hormesis phenomenon [39] characterized by a low dose stimulation producing a beneficial response, contrary to a high dose, which would have an inhibitory or toxic effect. In our literature search, we found no specific publication about hormesis due to carboplatin. However, this phenomenon is observed for cisplatin, which has a similar biological action as carboplatin [40].

### 2.2. In Vivo Antitumoral Efficacy

Figure 2 shows the tumor growth delay after four different treatments (carboplatin, GNP, GNP + carboplatin and LipoGold) at different dosages, with or without X-ray irradiation. The time to reach five times the initial volume of the tumor ±0.2 (5IT) was chosen as a reference point to compare each group. 5IT is also reported in Figure 3. The control group was treated with injection of phosphate buffer saline (column 3) with a 5IT of 9.9 days. No significant difference was observed between PBS (phosphate buffer saline) (column 3), the combination of carboplatin [0.72 µg] plus GNP [5.3 µg] (column 2) and LipoGold (column 4). Only a high dose of carboplatin [0.72 µg] (column 5) shows a small benefit (*p* < 0.05) compared to the PBS group. Interestingly, GNP alone (column 1) shown a faster tumor growth than PBS control (*p* < 0.05). This may be another hormesis phenomenon due to an unknown protective role of low dose of GNP on tumor cell. The same potential protective effect is also observable when we compare carboplatin [0.72 µg] (column 5) to the combination of carboplatin [0.72 µg] plus GNP [5.3 µg] (column 2) (*p* < 0.05). Treatment with radiation alone produces a 5IT of 40.6 days (column 8), but combination of radiation with carboplatin [0.72 µg] + 10 Gy (column 6) or the combination of carboplatin [0.72 µg] plus GNPs [5.3 µg] + 10 Gy (column 7) do not affect the 5IT. LipoGold (carboplatin [0.72 µg] plus GNP [5.3 µg]) + 10Gy (column 9) has a 5IT of 50.4 days. It takes a much higher dose of GNP alone [366.3 µg] than in LipoGold (column 10) to bring 5IT to 53.9 days.

The in vivo tumor growth delay curves of the HCT116 tumor cells implanted in NU/NU nude mice are shown in Figure 2. Each group were compared at the 5IT ± 0.2 (red horizontal highlight) and reported in Figure 3. Small but significant differences were observed for the GNP (column 1) and carboplatin (column 5) groups compared to the control group (column 3), whereas no significant differences were observed for carboplatin + GNP (column 2) and LipoGold (column 4). Similar to the clonogenic assays, when IR is combined to the treatment, low dose of carboplatin (column 6) or carboplatin + GNPs (column 7) do not show significant differences compare to IR alone (column 8). *However, the same low concentration of carboplatin + GNPs in the liposomal formulation a synergistic effect occurs, with IR (column 9)*. High concentration of GNPs + IR (column 10) also enhances radiation therapy. It is interesting to compare the group treated with a low dose of carboplatin + GNPs (column 7) to the LipoGold group (column 9), which received the same concentrations of carboplatin and GNPs in encapsulated form. When these agents were not given in a liposomal formulation, no additional effect appeared, when combined with radiation. On the other hand, the liposomal formulation combined with radiation produces a significant delay on tumor growth. This difference appears to be due to a possible simultaneous cellular incorporation of carboplatin and GNPs, promoted by the liposomal formulation, as opposed to the injection of free carboplatin and GNPs that does not favor colocation in the cell. When a high dose of GNPs is combined to IR (column 10) a 5IT of about 54.75 days is observed. It should be noted that this is a concentration 69 times higher than that of the GNPs in LipoGold (GNPs = [5.3 µg]). The difference in GNP concentration alone cannot explain the difference of only 4 days compared to LipoGold (5IT = 50.35 days).

The mechanistic of incorporation of encapsulated drugs into cells via liposomal transport has been investigated in the literature. Many authors suggest that liposomes are mostly internalized by endocytosis and transported via the endolysosomal pathway [41,42,43]. Other study suggest direct liposomal fusion to the cell membrane or phagocytosis mechanism [44]. Our hypothesis is that the liposomal formulation will deliver all its load in the same cell allowing cellular colocation of the carried molecules. Presently, we do not know how these pathways combined to provide the much higher efficiency of colocation. Nevertheless, we find the present results to be consistent with our previous in vacuo assays showing that the combination of cisplatin and GNP + IR produces much greater cluster DNA damage compared to cisplatin + IR or GNPs + IR [37].

### 2.3. Parameters Relevant to Clinical Applications

A resume of publications studying gold nanoparticles combined with radiation is shown in Table 1. Recently, by using about ten times more GNPs than the amount of GNPs contained in LipoGold, we have obtained impressive results with separated injections of GNPs and cisplatin combined with radiation [45]. However, different conditions (radiation doses, cisplatin dose, follow-up time of tumor volume) do not allow perfect comparison with our results. To be clinically relevant, we have evaluated the amount of GNPs of different in vivo studies [25,31,32,45,46,47] that would correspond to a 60 kg human (last line in Table 1). We found that between 7.14 mg and 20 g of GNPs would have to be used for treatments in patients. With LipoGold, a more realistic in vivo treatment of 1.3 mg of GNP could be delivered via convection-enhanced delivery (CED) to achieve comparable results.

## 3. Materials and Methods

### 3.1. GNP Production

GNP were produced and stabilized with a thiopronin coating as described by Templeton et al. [49]. Briefly, 0.1033 g of tetrachloroauric acid (Sigma-Aldrich, Oakville, ON, Canada) and 0.1267 g of *N*-(2-mercaptopropionyl) glycerin were dissolved in 11.67 mL of 6:1 methanol: acetic acid. A solution of NaBH_4_ (0.2 g in 5 mL) was added. The suspension was stirred for 45 min and the solvent was removed by vacuum, purified by dialysis, dried by lyophilization and resuspended in double distillated water to a final concentration of 100 µg/mL. A mean diameter of 2.8 nm for GNPs was measured by transmission electron microscopy (Hitachi H-7500 transmission electron microscope operating at 60–80 kV, 15 µA, 1 × 10^−7^ Torr). A TEM image of a typical GNP distribution can be found in the on-line Appendix A.

### 3.2. Liposomal Preparation

Liposomes were produced by the staggered herringbone microfluidic method [50] (NanoAssemblr, Precision NanoSystem, Vancouver, BC, Canada). About 40 different combinations of lipid compositions (50 µmol/mL), aqueous were tried: Lipid ratios and flow rates to determine the best combination for producing LipoGold, which was intended to have a diameter between 100 nm to 140 nm and low polydispersity (lower than 0.2). The ultimate version was produced with a cationic lipid ratio of DOTAP: DSPC: Cholesterol of 45: 6.5: 48.5 with a flow rate of 4 mL/min and an aqueous lipids ratio of 3:1 at 65 °C. The aqueous solution containing GNPs (20 mg/mL) and carboplatin (20 mg/mL) was dissolved in water, whereas the lipid solution (50 µmol/mL) was dissolved in ethanol. To remove any unincorporated therapeutic agent and alcohol, the liposomal solution was purified through a Sephadex^TM^ G-50 column and the cloudy fractions were pooled and purified in a dialysis bag (12–14 kD MWCO) for four days with regular changes of distillated water. Dynamic light scattering (DLS) and polydispersity measurements on the purified liposomes were carried on with a ZETAPALS-Zeta potential Analyzer (Brookhaven instruments, Holtsville, NY, USA). Both measurements were automatically performed by this instrument, from which we could read that the polydispersity was 0.199 ± 0.023. The liposome mean diameter of 134.33 ± 0.27 nm was determined by transmission electron microscopy (Hitachi H-7500 transmission electron microscope operating at 60–80 kV, 15 µA, 1 × 10^−7^ Torr). A LipoGold TEM image is available in the on-line Appendix A. No trace of ethanol was measured with the alcohol test strips method (ALCO SCREEN, Vista, CA, USA). The liposomal solution (LipoGold) has a final concentration of gold (0.533 mg/mL) and carboplatin (0.072 mg/mL), measured by ICP-MS (X Series II from Thermo Scientific, Waltham, MA, USA). The concentration of carboplatin was the maximum possible to be incorporate in the liposomal formulation with the stock solution (10 mg/mL). The LipoGold formulation, made of cationic lipids [51,52], should be cationic, but the exact Zeta potential was not measured in the present study.

### 3.3. Cell Culture

HCT116 colorectal carcinoma cell line was obtained from American Type Culture Collection (Manassas, VA, USA). The cells were maintained in Minimum Essential Medium (MEM) (Gibco, NY, USA) containing 10% fetal bovine serum (FBS), 2 mM glutamine, 1 mM Sodium-Pyruvate, 100 units/mL penicillin and 100 μM streptomycin at 37 °C in a humidified atmosphere containing 5% CO_2_.

### 3.4. In Vitro Clonogenic Assays

Triplicates of cells were seeded in completed MEM at a density of 1.0 × 10^4^ cells per well in 96-well plates and incubated for 24 h. Cells were incubated with various drug concentrations in MEM without FBS for 4 h at 37 °C. The cells were irradiated with a Therapax HF150T x-ray source operated at 10.4 mA and 100 kV. The beam was filtered to yield photons with a most probable energy of ~ 80 keV. The dose of 2 Gy was delivered via a cone applicator with a 10 cm diameter at the base and a source to wells distance of 25 cm. Immediately after, cells were rinsed with PBS, trypsinated and plated in P100 petri dishes with complete MEM and incubated at 37 °C, 5% CO_2_. Colonies were counted 7 days later.

### 3.5. In Vivo Antitumoral Activity

Animal experimentations were approved by the Université de Sherbrooke Animal Care and Use Committee (Approval number 235-10). For all procedures (implantation, chemotherapy and radiotherapy), mice were anesthetized with an intraperitoneal injection of ketamine/xylazine (87/13 mg/mL) at 1 mL/kg.

For tumor implantation, HCT116 cells were inoculated (2 × 10^6^ cells in 0.1 mL) subcutaneously (s.c.) into each rear flank area of outbred male NU/NU nude mice at 4–6 weeks of age (*n* = 3–5) (Charles River Laboratories, Saint-Constant, QC, Canada). The tumor volume was measured starting one week post-implantation and then biweekly until treatment. Tumor volumes were calculated with the formula: V (mm^3^) = π/6 × a (mm) × b^2^ (mm^2^), where a and b were the largest and smallest perpendicular tumor diameters, respectively.

Chemotherapy treatments began when tumor volume reached a range of 30–60 mm^3^. Ten µL of tested solution (carboplatin [72 µg/mL], GNP [36.63 mg/mL], PBS 1X, LipoGold (carboplatin [72 µg/mL], GNP [0.53 mg/mL]) were injected by convection-enhanced delivery (CED) directly into the tumor. Tumor measurements and volume calculations were done 3 times per week until reaching the endpoint of 1 cm^3^, followed by euthanasia of the mice. Mice were not monitored for weight considering the problem of the added weight of the growing tumor. However, compared to healthy mice, tumor-bearing mice showed no visual sign of toxicity (loss of activity and grooming, or additional stress) up to the endpoint. No other toxicity analysis was performed.

Tumors were irradiated (10 Gy) 24 h after chemotherapy treatment, using a Therapax HF150T operated at 13.2 mA and 150 kV, equipped with the 1 cm diameter applicator and filter #8 (0.2 mm of aluminum and 1 mm of copper) to yield photons with an average energy of 86 keV. This radiation dose was chosen, because by itself, it induces a partial tumor response, which allowed assessment of the combined effect of chemotherapy with radiation. X-rays of ~80 keV were chosen for their capacity to deliver large quantities of LEEs with a relatively large penetration in tissue. Radiation was applied to both sides of the rear flank, where the tumor was implanted. Note: Experiments using GNP [366.3 µg] were performed by our group and have been published recently [25]. Rough data form [25] were used in the present publication for comparison needs.

### 3.6. Statistical Analysis

Paired *t*-test adjusted by False Discovery rate (Benjamini-Hochberg) was used to compare each group. Significant difference was inferred by a *p* value less than 0.05. Statistical tables relative to Appendix A.

## 4. Conclusions

The preliminary results presented in this paper have allowed investigating the in vitro and in vivo biological response of a wide range of combination of GNPs, X-ray irradiation, carboplatin and a liposomal formulation of carboplatin and GNPs. Now that the most efficient combination has been identified, we hope that our results will incite more elaborate research with the most interesting product (e.g., LipoGold) to fully establish its clinical potential. Among others, such studies should include details on the cellular incorporation of drugs, precise measurements of Zeta potentials and a more quantitative analysis of toxicity, including measurements of body weight changes. We also point out that valuable biological information could be obtained from a more detail analysis of the hormesis phenomenon reported in the present study.

In this study, mice bearing a human colorectal tumor were treated with radiation alone and in various combination of Pt-CA and GNPs administered by CED (i.e., direct low flow injection into the tumor) [53]. The intra-tumoral CED method was chosen because of its capacity to allow for up to hundred times more GNP tumor uptake than intravenous injection [25]. The most effective radiotherapeutic treatment was obtained when carboplatin and GNPs were brought together to the tumor cells via a liposomal carrier prior to irradiation. According to literature, the rationale behind liposomal transportation of CAs is mainly due for their favorable properties, such as biocompatibility, side effect reduction, and preferential tumor accumulation, which are caused by the enhance permeability and retention effect [54]. The present study suggests that in addition to these advantages, liposomal encapsulation could serve to simultaneously co-inject CAs and GNPs to considerably increase the efficiencies of chemoradiation therapy. The conjectured cellular colocation of platinum drugs and GNPs due to their encapsulation seems advantageous to reach a high level of radiosensitization, when only low concentrations of these compounds can be tolerated by the patient. CED of radiosensitizers, such as LipoGold are expected to increase considerably the radiation dose and RBE at the tumor level, while reducing radiation damage to healthy cells. Although much more information is needed to explain the metabolic and cellular behavior of LipoGold, the present investigation serves as an example of the guidelines that can be provided by fundamental research on the action of IR with biomolecules for the development of new types of treatments in radio-oncology. More precisely, this work and previous literature shows that GNPs can be used for efficient radio-enhancement of superficial tumors (e.g., colorectal cancer). However, to administer a realistic gold concentration in eventual clinical applications, colocalization of metal nanoparticles with a radiosensitizer, such as a platinum drug, appears to be a promising avenue.

## 5. Patents

US Patent application PCT/CA10/00583 “Compositions Comprising a Radiosensitizer and Anti-Cancer Agent and Methods of Uses Thereof”.

## Figures and Tables

**Figure 1 ijms-21-04848-f001:**
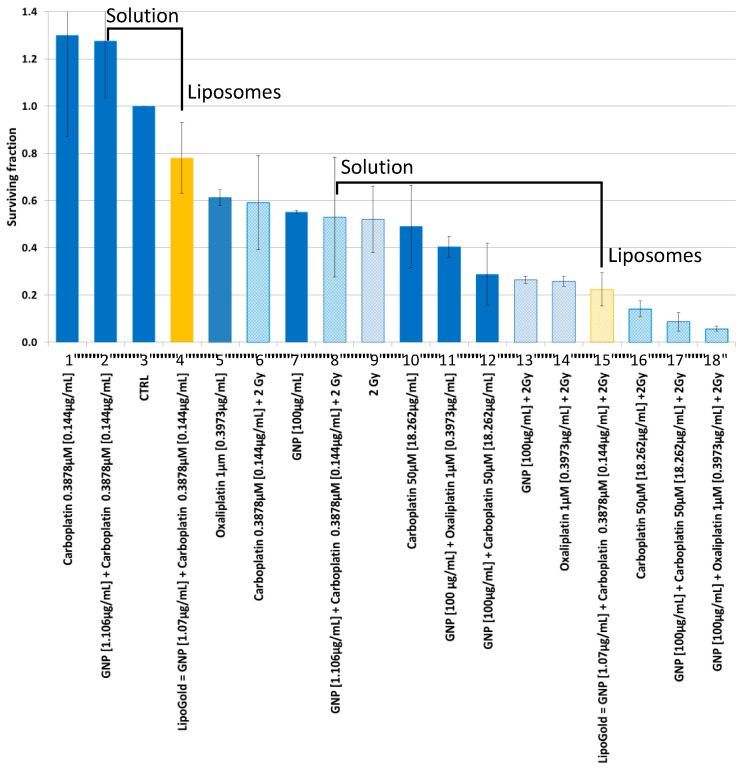
Clonogenic assay with a HCT116 colorectal cancer cell line. Survival fraction after different treatments combined or not with 2 Gy of X-ray radiation of ~80 keV. The measurements were repeated three times for each group. The efficiency of GNPs combined with carboplatin freely dissolved in solution or in liposomal formulation is highlighted.

**Figure 2 ijms-21-04848-f002:**
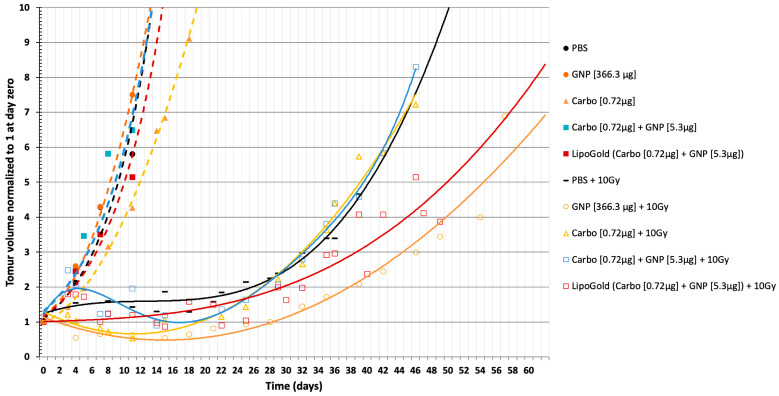
Tumor growth delay for different treatments of Nu/Nu nude mice bearing a subcutaneous HCT116 colorectal cancer cell line. The data passing through the pink horizontal strike-through trace at 5IT ± 0.2 are reported in Figure 3.

**Figure 3 ijms-21-04848-f003:**
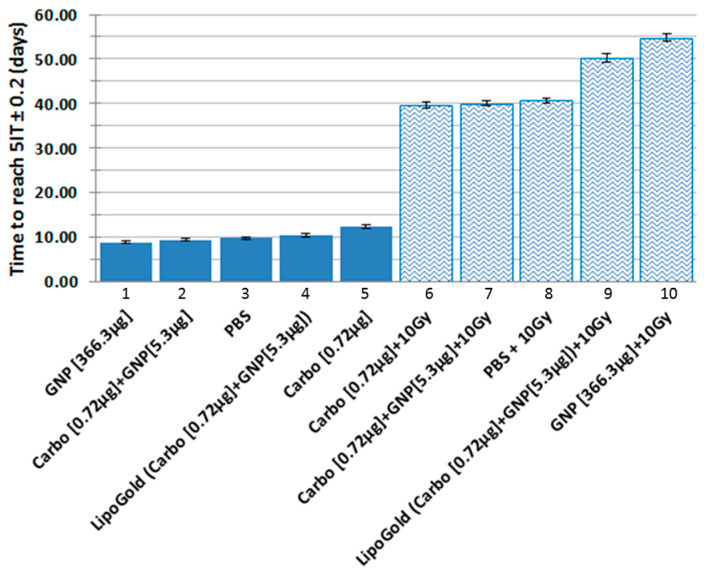
Time to reach five times the initial volume (5IT) after different treatments of Nu/Nu nude mice bearing a subcutaneous HCT116 colorectal cancer cell line. Data from the pink horizontal trace in Figure 2. The SD represent the surviving time passing through the pink line (5IT ± 0.2) of Figure 2.

**Table 1 ijms-21-04848-t001:** Comparison of key publications using gold nanoparticles (GNPs) for cancer treatment studies in relation to injected concentrations of GNPs.

Studies [Ref]	Hainfeld 2004 [32]	Chang 2008 [46]	Hainfeld 2013 [47]	Bobyk 2013 [31]	Shi 2016 [25]	Cui 2017 [45]	This Study	
Tumor type	Mammary carcinoma	Melanoma	Malignant glioma	Glioma	Colorectal cancer	Breast cancer	Colorectal cancer	
Animal model	Mice	Mice	Mice	Rats	Mice	Mice	Mice	
Methods	i.v.	i.v.	i.v.	Convection-enhanced delivery (CED)	Intra tumoral infusion	Intra tumoral infusion	CED (GNP)	CED (LipoGold)
GNP (mg/animal) ^1^	35	0.671	80	0.125 to 0.250	0.3663	0.05 to 0.5	0.3663	0.0053
Initial tumor volume (mm^3^)	50–90	50 to 90	14.13 to 65.42	33.5	100	250	60	60
Equivalent for human of 60 kg ^2^ (mg of AuNPs)	8750	167.74	20,000	7.14 to 14.29	91.57	12.5 to 125	91.57	1.325

^1^ Assuming weight of mouse and rat of 20 g and 150 g respectively. ^2^ According to Freireich, E.J. et al., 1966 [48].

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
