# Peer review of "Concomitant Chemoradiation Therapy with Gold Nanoparticles and Platinum Drugs Co-Encapsulated in Liposomes"

_ijms, 2020, doi:10.3390/ijms21144848_

Round 1

Reviewer 1 Report

The manuscript by Sanche et al. concerns the development of tumour therapy based on the application of gold nanoparticle-based radiosensitizers. In the manuscript, the authors have suggested (based on their previous observations) to use a combination of the gold nanoparticle-based radiosensitizers with the platinum-based chemotherapeutic agents to enhance the effectiveness of tumour therapy. Moreover, in the manuscript, a liposome-based carrier (a liposomal vehicle named LipoGold) has been developed to improve the delivery of the radiosensitizer to the tumour.

The cancer therapy is an acute problem of modern biomedicine. In this way, the study by Sanche et al. is quite actual, and can therefore be of interest to the readers.

Generally, I find the quality of the manuscript to be high. However, certain details of the experiments performed must be specified before the manuscript can be published. Also, despite the language of the manuscript is fine, the authors are encouraged to check the spelling carefully, just to eliminate several confusing misprints. Below, I would like to list my comments and suggestions to the authors.

Comments and suggestions to the authors:

  1. In the introduction, the authors have carefully reviewed the literature regarding the physical basis of the action of the gold nanoparticle-based radiosensitizers on the tumour. The studies considering the biological effects of these radiosensitizers, as well as the effects of the platinum-based chemotherapeutic agents, have also been properly reviewed and cited. In the very first paragraph of the introduction (on P. 1), the authors state the following: “A promising avenue to increase this dose, while decreasing collateral damage consists of developing radiosensitizers or radio-enhancers that increase relative biological effectiveness (RBE) at the tumor. In principle, such treatment could be optimized byinjecting the radiosensitizer directly into the tumor together with a chemotherapeutic agent. In this case, the detrimental effects of increased radiation dose and chemotherapy would be much more localized at the tumor.” Despite these statements are quite logical, the authors are encouraged to support these statements by citing the appropriate literature.
  2. The authors should expand and specify certain details of the experiments regarding the characterization of the obtained gold nanoparticles. The nanoparticles were obtained using a standard technique, and this is clearly stated in the Materials and Methods (Section 3.1). However, the description of the further transmission electron microscopy (TEM) and dynamic light scattering (DLS) experiments, performed to characterize the obtained nanoparticles, are quite incomplete. The DLS and TEM experiments must be described in detail, so that they could be reproduced. Neither the electron microscope nor the DLS device models are specified; this should be done. The TEM sample preparation procedure, as well as the measurement conditions, must be described in the manuscript. The same refers to the DLS measurements. If previously published techniques have been used throughout these measurements, the corresponding references can be cited alternatively. The typical TEM images and size distribution curves (obtained by DLS) of the synthesized gold nanoparticles should be provided either within the Materials and Methods section of the manuscript or as a supplementary information.

Also, polydispersity measurements are mentioned in Section 3.2. What method has been employed for this purpose? This should be specified, and the results obtained should be presented either within the manuscript or as a supplementary information.

  1. Several misprints should be corrected, namely:
  2. 52: Written: “…GNPs are essentially generators…” expected: “…GNPs are essential generators…”;
  3. 179: Written: “…is shows in Table…” expected: “…is shown in Table…”;
  4. 195: Subscript is expected in the NaBH4 formula;
  5. 196: Written: “dry by lyophilization” expected: “dried by lyophilization”;
  6. 197: Written: “…for a final concentration of…” expected: “…to a final concentration of…”;
  7. 198: Written: “by transmission electron microscope” expected: “by transmission electron microscopy”;
  8. 200: Written: “…with the staggered herringbone microfluidic method” expected: “…by the staggered herringbone microfluidic method”;
  9. 201: Written: “NanoAssemblr”. It seems like there is a misprint in the company name. Please check and correct if necessary;
  10. 242-243: Written: “Ten μL of carboplatin was injected…” expected: “Ten μL of carboplatin were injected…”;
  11. 246-247: Written: “…using a Therapax HF150T was operated at…” expected: “…using a Therapax HF150T operated at…”;
  12. 247-248: no end bracket within the sentence. Please correct to make it better understandable;
  13. 252-253: Written: “Experiments … were performed by our group and has been published recently” expected: “Experiments … were performed by our group and have been published recently”.

Author Response

Response to the reviewers

We would like to thank the three reviewers for their pertinent and helpful comments and questions. These definitively improved the presentation of our work. Corresponding changes were made in the manuscript, when applicable.  

Reviewer #1
Comments and suggestions to the authors:

  1. In the introduction, the authors have carefully reviewed the literature regarding the physical basis of the action of the gold nanoparticle-based radiosensitizers on the tumour. The studies considering the biological effects of these radiosensitizers, as well as the effects of the platinum-based chemotherapeutic agents, have also been properly reviewed and cited. In the very first paragraph of the introduction (on P. 1), the authors state the following: “A promising avenue to increase this dose, while decreasing collateral damage consists of developing radiosensitizers or radio-enhancers that increase relative biological effectiveness (RBE) at the tumor. In principle, such treatment could be optimized by injecting the radiosensitizer directly into the tumor together with a chemotherapeutic agent. In this case, the detrimental effects of increased radiation dose and chemotherapy would be much more localized at the tumor.” Despite these statements are quite logical, the authors are encouraged to support these statements by citing the appropriate literature.

    NOTE TO REVIEWER 1: Reviewer 1 is right. Sometimes it is so obvious to the writer that he forgets to support a statement with appropriate literature. References are now added at line 35 in the beginning of the introduction (Ref. 1-2).

  1. The authors should expand and specify certain details of the experiments regarding the characterization of the obtained gold nanoparticles. The nanoparticles were obtained using a standard technique, and this is clearly stated in the Materials and Methods (Section 3.1).

    However, the description of the further transmission electron microscopy (TEM) and dynamic light scattering (DLS) experiments, performed to characterize the obtained nanoparticles, are quite incomplete. The DLS and TEM experiments must be described in detail, so that they could be reproduced. Neither the electron microscope nor the DLS device models are specified; this should be done. The TEM sample preparation procedure, as well as the measurement conditions, must be described in the manuscript. The same refers to the DLS measurements. If previously published techniques have been used throughout these measurements, the corresponding references can be cited alternatively. The typical TEM images and size distribution curves (obtained by DLS) of the synthesized gold nanoparticles should be provided either within the Materials and Methods section of the manuscript or as a supplementary information.

NOTE TO REVIEWER 1: We must apologize for this mistake. DSL was used only for the characterization of the liposomal formulation. This is now indicated in lines 223-227. The gold nanoparticle size was measured by TEM only. Characteristics of the TEM are now added as requested (sections 3.1. lines 238-240).

Also, polydispersity measurements are mentioned in Section 3.2. What method has been employed for this purpose? This should be specified, and the results obtained should be presented either within the manuscript or as a supplementary information.

NOTE TO REVIEWER 1: Dynamic light scattering (DLS) and polydispersity measurements on the purified liposomes were carried on with a ZETAPALS-Zeta potential Analyzer (Brookhaven instruments, Holtsville, NY). Both measurements were automatically performed by this instrument, from which we could read that the polydispersity was 0.199 ± 0.023. The liposome mean diameter of 134.33 ± 0.27 nm was determined by transmission electron microscopy (Hitachi H-7500 transmission electron microscope operating at 60-80 kV, 15µA, 1 X 10-7 Torr). A LipoGold TEM image is available in the on-line Supplementary Material. This is now indicated in the text in Section 3.2 lines 253-259.

  1. Several misprints should be corrected, namely:

NOTE TO REVIEWER 1: The following misprints (2 to 13) were corrected, except #9 which is the right typography. We thank the referee for these corrections.

  1. 52: Written: “...GNPs are essentially ..” expected: “...GNPs are essential

generators...”;

  1. 179: Written: “...is shows in Table...” expected: “...is shown in Table...”;
  2. 195: Subscript is expected in the NaBH4 formula;
  3. 196: Written: “dry by lyophilization” expected: “dried by lyophilization”;
  4. 197: Written: “...for a final concentration of...” expected: “...to a final concentration of...”;
  5. 198: Written: “by transmission electron microscope” expected: “by transmission electron

microscopy”;

  1. 200: Written: “...with the staggered herringbone microfluidic method” expected: “...by the

staggered herringbone microfluidic method”;

  1. 201: Written: “NanoAssemblr”. It seems like there is a misprint in the company name. Please

check and correct if necessary;

  1. 242-243: Written: “Ten μL of carboplatin was ..” expected: “Ten μL of carboplatin

were injected...”;

  1. 246-247: Written: “...using a Therapax HF150T was operated ..” expected: “...using a

Therapax HF150T operated at...”;

  1. 247-248: no end bracket within the sentence. Please correct to make it better understandable;
  2. 252-253: Written: “Experiments ... were performed by our group and has been published

recently” expected: “Experiments ... were performed by our group and have been published recently”.

Reviewer 2 Report

The authors described usefulness of chemoradiotherapy combined with gold nanoparticles and platinum drugs co-encapsulated in liposomes. There are some scientific issues to resolve in this manuscript.

  1. The authors used liposome encapsulated with gold nanoparticles and carboplatin (LipoGold) and the size of the LipoGold was 134 nm in a diameter. In the DDS (drug delivery system) field, this size of drug was suitable for the EPR (enhanced permeability and retention) effect and the drug could be accumulated to the tumor. The authors should discuss why this liposome was injected directly into the tumor.
  2. Figure 1 was confused. Figure should be shown separately in the several conditions. Why did the authors use only 2 drug concentrations in vitro study?
  3. In the result section, the authors mentioned that carboplatin with or without GNP at very low doses could stimulate the cells proliferation. The authors should discuss why low dose anticancer drug induce cell proliferation.
  4. In vivo toxicity (such as body weight change or several organ toxicity) of each drug should be shown.
  5. The authors described that a high dose of carboplatin showed a small benefit (P < 0.05) compared to the PBS group. The authors should discuss why the combination group with carboplatin and GNP didn’t show the benefit.
  6. The authors discussed that the liposomal formulation combined with radiation produced a significant delay on tumor growth due to the simultaneous cellular incorporation of drugs. The internalization of the drugs should be shown in the figure.
  7. In the method section, the authors described that the LipoGold was measured by ZETAPALS-Zeta potential Analyzer. Please add a zeta potential of the LipoGold in the result.
  8. Paired t-test should not be used for multiple comparison.

Author Response

Response to the reviewers

We would like to thank the three reviewers for their pertinent and helpful comments and questions. These definitively improved the presentation of our work. Corresponding changes were made in the manuscript, when applicable.  

Reviewer #2

  1. The authors used liposome encapsulated with gold nanoparticles and carboplatin (LipoGold) and the size of the LipoGold was 134 nm in a diameter. In the DDS (drug delivery system) field, this size of drug was suitable for the EPR (enhanced permeability and retention) effect and the drug could be accumulated to the tumor. The authors should discuss why this liposome was injected directly into the tumor.
    NOTE TO REVIEWER 2: Reviewer #2 is right. Liposomal formulation allows for a preferential tumour uptake due by the EPR effect. However, in a previous publication, we have shown that the intra-tumoral CED method allows up to hundred time more GNP tumor uptake than intravenous injection [46]. We have kept this method to avoid introducing variable for the comparison between the liposomal formulation and the other products used in this study. An explanation is now inserted in the Conclusion section, lines 366-367.

  2. Figure 1 was confused. Figure should be shown separately in the several conditions. Why did the authors use only 2 drug concentrations in vitro study?
    NOTE TO REVIEWER 2: The concentrations of carboplatin used in this experiment represent two pertinent concentrations which are the EC50 [50µM] of for this cell line and the maximum concentration [0.3878µM] that we were able to incorporate into the liposomal formulation at this time. This is now mentioned at line 262-264.

  3. In the result section, the authors mentioned that carboplatin with or without GNP at very low doses could stimulate the cells proliferation. The authors should discuss why low dose anticancer drug induce cell proliferation.
    NOTE TO REVIEWER 2: Our hypothesis about the increase cell proliferation when they are exposed to low dose of carboplatin is that it could be possible that non-lethal dose of carboplatin trigger expression of repair system that promotes cell survival. This hypothesis is now mentioned at the end of the section 2.1, lines 134-135

  4. In vivo toxicity (such as body weight change or several organ toxicity) of each drug should be shown.
    NOTE TO REVIEWER 2: We agree that such measurements would be significant in evaluating the future applicability of the various combinations. Unfortunately, such measurements were not considered in the present study. As we mentioned in lines 90-92 “Preliminary results are concisely presented in this article on this novel method of combining metal nanoparticles with CAs in an effort to enhance the benefits of concomitant chemoradiation therapy”. Certainly, more information would be valuable, but we hope to generate with these results sufficient interest in the field, to encourage further work on LipoGold, as well as funding of future experiments.

  5. The authors described that a high dose of carboplatin showed a small benefit (P < 0.05) compared to the PBS group. The authors should discuss why the combination group with carboplatin and GNP didn’t show the benefit.
    NOTE TO REVIEWER 2: This is an interesting observation. We don’t know the reason for this lack of benefit. A possible hypothesis could be that low dose of GNP may protect the tumor cells by an unknown mechanism. This is mentioned in section 2.2., line 149-153.

  6. The authors discussed that the liposomal formulation combined with radiation produced a significant delay on tumor growth due to the simultaneous cellular incorporation of drugs. The internalization of the drugs should be shown in the figure.
    NOTE TO REVIEWER 2: Unfortunately, the cellular internalization of GNPs and carboplatin was not measure in this study. The exact statement in the text was : “This difference appears to be due to the simultaneous cellular incorporation …” Section 2.2., line 184

    To avoid any confusion, the following changes were made.

Line 182-186, Section 2.2 :  “This difference appears to be due to a possible simultaneous cellular incorporation of carboplatin and GNPs, promoted by the liposomal formulation, as opposed to the injection of free carboplatin and GNPs that does not favor colocation in the cell”.

Line 193-195, Section 2.2 : “Our hypothesis is that the liposomal formulation will deliver all its load in the same cell allowing cellular colocation of the carried molecules.” -

Line 367-369, Conclusion : “The most effective radiotherapeutic treatment was obtained when carboplatin and GNPs were incorporated brought togehter in to the tumor cells via a liposomal carrier prior to irradiation

Line 374-376, Conclusion : “The conjectured cellular colocation of platinum drugs and GNPs…”

  1. In the method section, the authors described that the LipoGold was measured by ZETAPALS- Zeta potential Analyzer. Please add a zeta potential of the LipoGold in the result.
    NOTE TO REVIEWER 2: Dynamic light scattering and polydispersity were measured using the apparatus ZETAPALS- Zeta potential Analyzer. Zeta potential itself wasn’t measured. Lines 253-259 in section 3.2 are now restructured for more clarity.              

  1. Paired t-test should not be used for multiple comparison.
    NOTE TO REVIEWER 2: Reviewer #2 is right. However, as shown in the Supplementary Material, the statistics relative to data from Figures 1 and 2 were done 2 groups at a time.

Reviewer 3 Report

The authors test the in vitro and in vivo radiosensitizing effect of a combination of the carboplatin and GNPs against tumor cells. The manuscript can be accepted for publication in International Journal of Molecular Sciences, after minor revision. The authors should revise the manuscript according to the following comments.

1.The part of the characterization of the liposomes should be transferred in the results and discussion
2. The authors should explain in the introduction part why they choose to test human colorectal cancer cells
3. The authors should explain what the survival fractions are representing in the text
4. Give the meaning of the abbreviation IR
5. The size of the figure 1 is huge, moreover, it is difficult to be read the labels and the numbering of the columns
6. Please rephrase the heading "Keeping in Mind Relevant Treatments for Clinical Application"
7. The rows of the table 1 should ne in one line
8. The authors mention in the experimental part that " Ten μL of carboplatin was injected by convection-enhanced delivery (CED) directly into the tumor" the authors should give the concentration of the carboplatin.
Grammar errors
a. "cytotoxicicity"

Author Response

Response to the reviewers

We would like to thank the three reviewers for their pertinent and helpful comments and questions. These definitively improved the presentation of our work. Corresponding changes were made in the manuscript, when applicable.  

Reviewer #3

1.The part of the characterization of the liposomes should be transferred in the results and discussion.
NOTE TO REVIEWER 3: Considering that this article is mainly about the biological effect on tumour of combined chemotherapy, gold nanoparticles and radiation rather than liposomal production, we prefer to keep the liposomal characterisation in the Materials and Methods section. 

  1. The authors should explain in the introduction part why they choose to test human colorectal cancer cells
    NOTE TO REVIEWER 3: Justification of the choice of colorectal cancer model is now present in the text toward the end of the introduction, lines 88-90).

  2. The authors should explain what the survival fractions are representing in the text
    NOTE TO REVIEWER 3:The following explanation is now added at the beginning of section 2.1, lines 104-105) The mean survival fraction (SF) for all groups are relative to the control group (CTRL, column 3), which is normalized to 1.
  3. Give the meaning of the abbreviation IR
    NOTE TO REVIEWER 3: The definition of IR is present at its first mention in the introduction line 53-54 and in the Abbreviation section at line 420.

  4. The size of the figure 1 is huge, moreover, it is difficult to be read the labels and the numbering of the columns.
    NOTE TO REVIEWER 3: The image quality is very clear on our side. It is possibly due to a file conversion. We will take an attentive care of this issue in the final approval of the manuscript.

  5. Please rephrase the heading "Keeping in Mind Relevant Treatments for Clinical Application"

NOTE TO REVIEWER 3: We changed the heading of Section 2.3 to “Parameters Relevant to Clinical Applications” and that of Table 1 to “Comparison of key publications using Gold Nanoparticles for cancer treatment studies in relation to injected concentrations of GNPs”.

  1. The rows of the table 1 should ne in one line
    NOTE TO REVIEWER 3: Revised as requested. Particular attention will be taken in the version to be printed.

  2. The authors mention in the experimental part that " Ten μL of carboplatin was injected by convection-enhanced delivery (CED) directly into the tumor" the authors should give the concentration of the carboplatin.
    NOTE TO REVIEWER 3: Section 3.5 lines 345-347, revised as requested and also for the other solutions.

Grammar errors
a. "cytotoxicicity" NOTE TO REVIEWER 3: revised as requested

Round 2

Reviewer 2 Report

The authors replied to my comments; however, there are still some scientific issues to resolve in this manuscript.

  1. The authors used liposome encapsulated with gold nanoparticles and carboplatin (LipoGold) and the size of the LipoGold was 134 nm in a diameter. In the DDS (drug delivery system) field, this size of drug was suitable for the EPR (enhanced permeability and retention) effect and the drug could be accumulated to the tumor. The authors should discuss why this liposome was injected directly into the tumor.

NOTE TO REVIEWER 2: Reviewer #2 is right. Liposomal formulation allows for a preferential tumour uptake due by the EPR effect. However, in a previous publication, we have shown that the intra-tumoral CED method allows up to hundred time more GNP tumor uptake than intravenous injection [46]. We have kept this method to avoid introducing variable for the comparison between the liposomal formulation and the other products used in this study. An explanation is now inserted in the Conclusion section, lines 366-367.

→OK, I can agree.

  1. Figure 1 was confused. Figure should be shown separately in the several conditions. Why did the authors use only 2 drug concentrations in vitro study?

NOTE TO REVIEWER 2: The concentrations of carboplatin used in this experiment represent two pertinent concentrations which are the EC50 [50µM] of for this cell line and the maximum concentration [0.3878µM] that we were able to incorporate into the liposomal formulation at this time. This is now mentioned at line 262-264.

→I can understand why the authors use only 2 drug concentration. However, Figure 1 was still confused for me. What was most important thing? Which columns should be compared?

  1. In the result section, the authors mentioned that carboplatin with or without GNP at very low doses could stimulate the cells proliferation. The authors should discuss why low dose anticancer drug induce cell proliferation.

NOTE TO REVIEWER 2: Our hypothesis about the increase cell proliferation when they are exposed to low dose of carboplatin is that it could be possible that non-lethal dose of carboplatin trigger expression of repair system that promotes cell survival. This hypothesis is now mentioned at the end of the section 2.1, lines 134-135

→If the authors describe that low dose of carboplatin promotes cell proliferation, the authors should discuss more with several scientific papers.

  1. In vivo toxicity (such as body weight change or several organ toxicity) of each drug should be shown.

NOTE TO REVIEWER 2: We agree that such measurements would be significant in evaluating the future applicability of the various combinations. Unfortunately, such measurements were not considered in the present study. As we mentioned in lines 90-92 “Preliminary results are concisely presented in this article on this novel method of combining metal nanoparticles with CAs in an effort to enhance the benefits of concomitant chemoradiation therapy”. Certainly, more information would be valuable, but we hope to generate with these results sufficient interest in the field, to encourage further work on LipoGold, as well as funding of future experiments.

→I cannot agree. The toxic analysis (at least body weight change) is important for animal study. Since the authors mentioned this result was preliminary, the confirmation or validation test with toxicity was needed.

  1. The authors described that a high dose of carboplatin showed a small benefit (P < 0.05) compared to the PBS group. The authors should discuss why the combination group with carboplatin and GNP didn’t show the benefit.

NOTE TO REVIEWER 2: This is an interesting observation. We don’t know the reason for this lack of benefit. A possible hypothesis could be that low dose of GNP may protect the tumor cells by an unknown mechanism. This is mentioned in section 2.2., line 149-153.

→I think a high dose of carboplatin group may be no significant difference compared to PBS group analyzed by a multiple comparison test. In the scientific paper, the authors should mention scientific comments. I cannot agree “unknown mechanism”.

  1. The authors discussed that the liposomal formulation combined with radiation produced a significant delay on tumor growth due to the simultaneous cellular incorporation of drugs. The internalization of the drugs should be shown in the figure.

NOTE TO REVIEWER 2: Unfortunately, the cellular internalization of GNPs and carboplatin was not measure in this study. The exact statement in the text was : “This difference appears to be due to the simultaneous cellular incorporation …” Section 2.2., line 184

To avoid any confusion, the following changes were made.

Line 182-186, Section 2.2 : “This difference appears to be due to a possible simultaneous cellular incorporation of carboplatin and GNPs, promoted by the liposomal formulation, as opposed to the injection of free carboplatin and GNPs that does not favor colocation in the cell”.

Line 193-195, Section 2.2 : “Our hypothesis is that the liposomal formulation will deliver all its load in the same cell allowing cellular colocation of the carried molecules.” -

Line 367-369, Conclusion : “The most effective radiotherapeutic treatment was obtained when carboplatin and GNPs were incorporated brought togehter in to the tumor cells via a liposomal carrier prior to irradiation

Line 374-376, Conclusion : “The conjectured cellular colocation of platinum drugs and GNPs…”

→I can understand the measurement of the internalization into cells or colocation on cells of GNP or carboplatin is difficult to assess in vivo and the authors mention that these drugs were “incorporated” not “internalized”. But I think the authors can show the drug concentration of these drug in the tumor at 24 hr after drug injection (same time as irradiation).

  1. In the method section, the authors described that the LipoGold was measured by ZETAPALS- Zeta potential Analyzer. Please add a zeta potential of the LipoGold in the result.

NOTE TO REVIEWER 2: Dynamic light scattering and polydispersity were measured using the apparatus ZETAPALS- Zeta potential Analyzer. Zeta potential itself wasn’t measured. Lines 253-259 in section 3.2 are now restructured for more clarity.            

→I can understand. However, Zeta potential as well as particle size was important information for liposome. Thus, I think the readers want to know whether the LipoGold is cationic or anionic. If the authors reported previously, please add the information.

  1. Paired t-test should not be used for multiple comparison.

NOTE TO REVIEWER 2: Reviewer #2 is right. However, as shown in the Supplementary Material, the statistics relative to data from Figures 1 and 2 were done 2 groups at a time.

→I cannot agree. These results were shown in same figure, and analyzed at same time and by same method. Multiple comparison, such as Dunnett or Tukey test, must be used in these cases.

Author Response

We would like first to thank Reviewer #2 for his additional constructive comments and suggestions. We fully agree with this reviewer that that more information on our experiments, as suggested in his comments 4, 6 and 7, would increase the quality of our manuscript. We have done our best in the revised version to comply with the referee suggestions, and believe that this version is considerably improved, due to his comments and suggestions. However, we would like to mention that to do all required measurements to fully comply with all suggestions is not possible presently. Let me explain. This study was carried on many years ago, while I was working in Pr. Sanche laboratory. The samples that were produced at this time are no longer available, nor can they be reproduced in our faculty. This multidisciplinary research was part of global effort involving other groups, particularly with expertise in liposomal encapsulation. Moreover, I am presently working in another laboratory, on a much different project. The same situation is to be found with our partners in British Columbia, Canada. I fully agree that the more information we can get the better is the work and the manuscript. It is for this reason that we waited so many years before publishing these results, hoping to get together again with our previous partners and perform more detailed investigations. Such experiments are costly and involve travelling and we have not been able, so far, to get back together. We finally decided to submit for publication our present data, as preliminary results. We hope that publishing this paper will interest other groups, attract appropriate funding to continue our studies in more details, allow us to collaborate again with other groups and provide information, beyond that requested by the referee.

Round 3

Reviewer 2 Report

I can understand the authors' situation. I can agree this study is preliminary data and future collaborative study will be continued according to these results. I accepted the authors’ comments.